# Antibacterial Effect of 16 Essential Oils and Modulation of *mex* Efflux Pumps Gene Expression on Multidrug-Resistant *Pseudomonas aeruginosa* Clinical Isolates: Is Cinnamon a Good Fighter?

**DOI:** 10.3390/antibiotics12010163

**Published:** 2023-01-12

**Authors:** Răzvan Lucian Coșeriu, Camelia Vintilă, Mirela Pribac, Anca Delia Mare, Cristina Nicoleta Ciurea, Radu Ovidiu Togănel, Anca Cighir, Anastasia Simion, Adrian Man

**Affiliations:** 1Department of Microbiology, George Emil Palade University of Medicine, Pharmacy, Science and Technology Târgu Mureș, 540142 Târgu Mureș, Romania; 2Doctoral School, George Emil Palade University of Medicine, Pharmacy, Science and Technology Târgu Mureș, 540142 Târgu Mureș, Romania; 3Nutrition & Holistic Health, Holomed, 540272 Târgu Mureș, Romania

**Keywords:** essential oils, *Pseudomonas aeruginosa*, cinnamon, efflux pumps, gene expression

## Abstract

The purpose of the study was to describe the antimicrobial activity of 16 common essential oils (EOs) on multidrug-resistant (MDR) *Pseudomonas aeruginosa* clinical isolates, including the determination of the effects on *mex* efflux pumps gene expression. Seventy-two clinical isolates of *P. aeruginosa* collected between 2020–2022 were screened for susceptibility to EOs using Kirby–Bauer disk diffusion to identify potential candidates for future alternative therapies. The minimal inhibitory concentration (MIC) was further determined for the EO that proved antibacterial activity following the disk diffusion screening. Positive and negative controls were also used for method validation. Since cinnamon EO exhibited the best antimicrobial activity, it was further used to evaluate its influence on *mex* A, B, C, E, and X efflux pumps gene expression using real-time RT-PCR. Cinnamon EO inhibited all *P. aeruginosa* strains, followed by thyme EO (37.5%, *n* = 27) and lavender EO (12.5%, *n* = 9). The other EOs were less efficient. The MIC detection showed that cinnamon at a concentration of 0.05% *v*/*v* inhibited all MDR *P. aeruginosa* isolates. Thyme, turmeric, peppermint, basil, clove, and lavender EOs presented various results, most of them having activity at concentrations higher than 12.5% *v*/*v*. By studying the activity of cinnamon EO on *mex* efflux pumps, it was found that *mex*A and *mex*B (66.5%) were generally under-expressed. The remarkable results produced using the very low concentrations of cinnamon EO, with 100% antimicrobial activity against multi-, extended-, and pan- drug-resistant (MDR, XDR, PDR) *P. aeruginosa* clinical isolates, completed with the severe alteration of the RNA messaging system, supports its potential to be used as adjuvant treatment, with impact on therapeutic results.

## 1. Introduction

*Pseudomonas aeruginosa*, a nosocomial bacteria responsible for respiratory tract infections, urogenital infections, and many other infections in Intensive Care Units [1], has rapidly evolved by acquiring complex resistance mechanisms. From soil to hospital wastewater systems, *P. aeruginosa*’s survival mechanisms include the creation of biofilms that allow it to populate spaces like toilets, sinks, or taps, being easily carried by hand from place to place. Influenced by the misuse/overuse of antibiotics and improper treatments, this highly adaptable bacteria managed to develop multidrug resistance (MDR) [2]. Moreover, MDR *P. aeruginosa* is not only found in hospitals but also in the environment [3]. *P. aeruginosa* is described as bacteria that have the potential risk of developing resistance to all known antibiotics [4,5].

In the era of extended antibiotic resistance, a class of antibiotics has been successfully implemented in the treatment of severe and difficult pathologies: the carbapenems. These inhibitors of cell wall synthesis are considered the last-line treatment option in situations where other antibiotics cannot be used anymore. However, the reversed situation happened not long after the use of these drugs: in 2013, the Centers for Disease Control (CDC) announced, as a priority, the need for carbapenem-resistance monitoring, especially the CPE (Carbapenems Producing Enterobacteriaceae) [6]. Bacteria managed to rapidly adapt to this new class of antibiotics through a diversity of mechanisms: enzyme production, alteration of membrane permeability, or mutation in efflux pumps [7].

Based on their hydrolyzation mechanisms, the carbapenemases have been classified into four Ambler groups: A, B, C, and D [8]. The production of carbapenemases is currently detectable using disk-diffusion methods, using commercial synergism tests, or using enzyme inactivation-based tests such as the Carba NP© test or carbapenem inactivation method [9]. A drawback of these tests is that they cannot detect other resistance mechanisms to carbapenems, which are not based on enzymatic activity, such as efflux pumps.

By combining an inner membrane transporter, an outer-membrane channel, and a periplasmic adapter protein, bacteria developed a highly active resistance mechanism called an efflux pump, which is able to efficiently transport antimicrobial agents out of the bacterial cells [10]. There are five families that describe and classify the efflux pumps. Four of them, the multidrug and toxic compound extrusion (MATE), the major facilitator (MF), the resistance nodulation-division (RND), and the small multidrug resistance or staphylococcal multi-resistance (SMR), use proton moving force for transportation of antimicrobial agents out of the cells. The fifth one, the ATP binding cassette (ABC), uses ATP (adenosine-triphosphate) energy [11,12,13]. These types of proteins are mostly encoded in the genome, but they were found both in plasmids and other transmissible elements [14]. The most common efflux pump family described in *Pseudomonas* spp. is RND. The efflux pumps are formed by a combination of OprM, which is found in the outer membrane, with MexA and MexB to form a stable complex at the inner membrane. In addition to MexA-B, OprM is capable of functional interactions with various RND/MF transporter complexes in *P. aeruginosa* [11,15,16]. Unlike the enzymatic activity of carbapenemases, the activity of efflux pumps is difficult to prove using common laboratory methods. Moreover, due to the involvement of combined resistance mechanisms, the common carbapenemase detection tests can be easily misinterpreted. The activity of *mex* efflux pumps can be phenotypically detected using Western Blot or Northern Blot assays, but they are considered elaborate, expensive, and time-consuming techniques, not accessible to the clinical laboratory [17]. The mutations in the bacterial genome play an important role in the overexpression of *mex* efflux pumps, which, aside from other resistance mechanisms, often lead bacteria to adapt to multiple classes of antibiotics. For example, by overexpressing the mexAB-OprM complex, bacteria can gain resistance to cephalosporins, penicillins, carbapenems, phenicols, and most fluoroquinolones [18].

*Pseudomonas* is one of the bacterial species that has always managed to develop resistance to a multitude of antibiotic classes, and researchers do their best to keep up with the discovery and development of new antimicrobial compounds [19,20]. Despite all efforts, the speed of the accommodation of bacteria to antimicrobial substances is much higher than the rate of research [21]. Therefore, there are situations where modern medicine has no alternative and where traditional medicine offers the option of using aromatic plants, known for their therapeutic properties that have improved symptoms of many illnesses for centuries [22]. These aromatic plants can be used in the form of dry plant material for infusions, capsules or tablets, tinctures, or EOs [23].

EOs can be defined as secondary metabolites of plants with a complex mixture of volatile compounds (mainly terpenes and hydrocarbons) [24,25,26]. The number of molecules and the chemical structures are highly diverse in EOs, with an average of 60 constituents in different concentrations [27].

Some EOs have proven antimicrobial activity, and this topic has become of high interest [28,29]. The main objective of this study was to determine the antimicrobial activity of 16 common EOs on carbapenem-resistant *P. aeruginosa* clinical isolates, which would support the potential use of these natural compounds in antimicrobial control. The secondary objective was to assess the modulation of antibiotic efflux-pump activity following exposure to MIC concentration of EOs, which could increase bacterial susceptibility to antibiotics.

## 2. Results

From the 160 strains of *P. aeruginosa* stoked during 2020–2022 in our laboratory, 72 strains corresponded to the inclusion criteria of highly increased resistance. Of these, 73.62% (*n* = 53) were considered XDR, 19.44% (*n* = 14) were MDR, and 6.94% (*n* = 5) were PDR, including colistin. All the strains were resistant to Meropenem 10 μg with an average diameter of 14.26 mm (SD-standard deviation = 8.73 mm).

The primary chemical components of the 14 EOs are presented in Table 1, according to the HPLC (High-Performance Liquid Chromatography) analysis that was performed by the producer during the quality control protocol.

### 2.1. Antimicrobial Activity of EOs

#### 2.1.1. Disk Diffusion Method

Cinnamon EO showed an inhibitory effect on all 72 clinical isolates (100%), with diameters over 10 mm, with an average of 24.72 mm (SD = 6.28), as exemplified in Figure 1. For the rest of the EOs, various results were obtained: *P. aeruginosa* presented susceptibility to thyme (37.5% of all isolates; *n* = 27), clove (8.33%; *n* = 6), lavender (12.5%; *n* = 9), basil (8.33%; *n* = 6), peppermint (5.55%; *n* = 4), and turmeric (1.38%; *n* = 1). All the other EOs failed to prove any antibacterial effect on all *P. aeruginosa* isolates, showing no inhibition zone following the disk-diffusion method.

The average diameter value obtained using Kirby–Bauer disk diffusion for meropenem was 14.26 mm (SD = 8.73 mm), consistent with the resistance to carbapenems according to the EUCAST (The European Committee on Antimicrobial Susceptibility Testing) standard. Cinnamon EO exerted a significantly better effect than meropenem (23.7 mm, SD = 4.74 mm, *p* < 0.05, CI 95%), proving the superior inhibitory effect of cinnamon. Thyme showed diameters quasi-similar with meropenem (14.47 mm, SD = 11.39 mm) but without statistical significance (*p* = 0.902, CI 95%).

#### 2.1.2. Minimum Inhibitory Concentration (MIC)

For the 7 EOs that, according to the Kirby–Bauer method, exerted inhibitory activity on the *P. aeruginosa* clinical isolates (cinnamon, thyme, turmeric, peppermint, basil, clove, lavender), the inhibitory (MIC) and bactericidal (MBC) effects were also tested. The results show that besides cinnamon, which presented the best MIC, all other EOs presented MIC values of <1.56% *v/v* on some strains, which were also consistent with good antimicrobial activity. Nevertheless, not all clinical isolates of *P. aeruginosa* responded in the same way; for example, compared to cinnamon EO (which efficiently inhibited all strains at very low MIC values), the other 6 EOs presented activity only on a few isolates, which were efficiently inhibited at low concentrations (<3% *v*/*v*). As presented in Table 2, some EOs showed inhibitory activity at a MIC of 12.5–25% (clove, peppermint, thyme), which makes them difficult to be used in vivo. MBC values were equal to the MIC values for 86.12% of the *P. aeruginosa* isolates and double the MIC for 13.88%. Details on MIC and MBC values are presented in the Appendix A.

Seven tested EOs (*Santalum paniculatum*, *Origanum vulgare*, *Origanum majorana*, *Rosmarinus officinalis*, *Pogostemon cablin*, *Melaleuca alternifolia*, *Eucalyptus citriodora*) did not show any inhibitory activity on *P. aeruginosa* isolates. By comparing their inhibitory activity to meropenem, no significant differences were found (*p* > 0.05, CI 95%). Also, no antimicrobial effects were exerted by any of the combined formulations OnGuard© and DDR Prime©.

### 2.2. Gene Expression of the Efflux Pumps

As the best antibacterial activity was shown by cinnamon EO, it was further used to evaluate its activity on the gene expression of efflux pumps. For this, the RNA obtained after extraction from *P. aeruginosa* strains with and without cinnamon EO was first quantified using nanodrop reading. It was found that the cinnamon EO significantly altered the average level of RNA in *P. aeruginosa* isolates (27.01 ng/μL; SD = 14.30, *p* = 0.01, CI 95%) compared to the average RNA level without exposure to the EO (43.44 ng/μL, SD = 18.75) (Figure 2).

Regarding the evaluation of gene expression using RT-PCR, an average Ct (cycle threshold) value of 26.37 (SD 5.43) was obtained without cinnamon EO and 31.86 (SD 5.39) with exposure to cinnamon EO (*p* < 0.05, CI 95%).

By comparing the average Ct obtained, as presented in Figure 3, the increased Ct values for samples incubated in the presence of cinnamon EO suggests inhibition of the activity of the efflux pumps. Significant differences were found for *mex* A, B, C, and E (*p* < 0.05, CI 95%) and not for *mex*X (*p* = 0.16, CI 95%).

Nevertheless, after evaluating the Fold Change (Fc) using ΔΔCt normalization against the housekeeping gene, both inhibition and stimulation activities were observed. We have to mention that the housekeeping gene *rpo*D was also under-expressed (*p* < 0.05, CI 95%), making the ΔΔCt calculations highly variable. For example, following normalization against *rpo*D, the best inhibitory activity of cinnamon EO was noticed on *mex*A and *mex*B, which were under-expressed in 66.7% (*n* = 10) of the *P. aeruginosa* tested isolates (Fc between 0.15–0.58); contrarily, *mex*C, respectively *mex*E and *mex*X have been over-expressed using cinnamon EO, for more than 86.7% (*n* = 13) and 66.7% (*n* = 10), respectively, of tested isolates.

All this data suggests that overall, cinnamon EO alters the gene expression in *P. aeruginosa*, which could be one of the factors that determine bacterial death.

## 3. Discussion

There is a clear unfavorable evolution of the resistance profile of *P. aeruginosa*, as presented in previous studies. The statistical data presented by ECDC (European Centre for Disease Prevention and Control) in 2019 reported that 3.4% of *P. aeruginosa* have high resistance profiles to five antimicrobial classes. In Romania, more than half of *P. aeruginosa* isolates were reported as resistant to carbapenems, showing an ascendant trend from 2015 to 2019 [30]. A previous study conducted by our institution in the period 2017–2022 also presented increased antibiotic resistance for *P. aeruginosa*, with 52.6% presenting resistance to imipenem, 42.2% to meropenem, and 56.3% to levofloxacin [31]. The presence of MDR *P. aeruginosa* has been described all over the world, with a variable prevalence of 25–50% [32,33,34,35]. Alarming results regarding MDR *P. aeruginosa* were presented in studies from the Middle East and Egypt (50–80%) [36]. This study describes, among the carbapenem-resistant *P. aeruginosa*, an increased proportion of XDR strains (73.62%) but also a small but clinically and epidemiologically significant number of PDR strains. All this data suggests a high variability of drug resistance among different geographic areas, which could be due to the different therapeutic protocols, compliance with microbial spreading limitation rules, or simply because the reporting of the isolates to superior institutions is not well managed. As it is recommended by CDC and ECDC or presented in clinical studies, the rational use of antibiotics is always emphasized [37,38,39]. This type of behavior should be implemented around the globe to prevent the selection of MDR/XDR *P. aeruginosa*, but not only those.

Considering the fact that bacteria manage to easily and rapidly develop resistance mechanisms to antibiotics, it is necessary to promote research for new alternatives that bring support to modern medical solutions. The interest in discovering new and natural plant extracts as antimicrobial agents has increased over the past years. There is increased interest in bioactive compounds provided by plants, such as EOs [40].

The term “essential oil” is derived from the drug Quinta essentia [41,42], assumed to originate from a statement credited to Swiss physician Phillippus Aureolus Theophrastus Bombastus von Hohenheim, also known as Paracelsus, who named the active component of a medicine mix “quinta essentia” [43]. An “essential oil” is defined by the International Standard Organization ISO 9235:2013 as a “product obtained from natural raw material of plant origin.” Most authors define essential oils as fragrant substance products or mixtures or as fragrant and odorless substance mixtures. In normal circumstances, these aromatic chemicals are chemically pure volatile molecules [44]. From our point of view, a more accurate scientific definition would be: “Essential oils are products or mixtures of products, which are formed in the cytoplasm and are normally present in the form of tiny droplets between cells. They are volatile and aromatic” [45]. This definition would further emphasize the biological origin of EOs. It should be mentioned that hundreds of years before the discovery of penicillin, treatments were generally based on plant extracts and EOs; therefore, we should not forget or ignore their potential today. At low concentrations, EOs exhibit scientifically proven advantages for in vivo use without toxic effects on human tissues [46,47].

Due to the hydrophobicity of EOs, a surfactant must be used for the emulsification. In our study, following a series of experiments prior to this study, Tween20 proved to have the best solubilizing activity for EOs, even if many researchers use DMSO (dimethyl sulfoxide) [48]. The EO emulsions present better antioxidant activity than the EOs alone and prevent the evaporation of volatile components from the culture medium [49]. Moreover, we have also decided to incubate at 50 °C [50] and sonicate the EO and Tween20, leading to a better homogenization of the EO with water. This is important from a methodological point of view because, for the detection of the MIC, the dilution of EO in the aqueous culture medium has to be reliable.

A large number of studies showed positive results regarding the antimicrobial activity of EOs, but most of them were performed on *P. aeruginosa* ATCC (American Type Culture Collection) standardized strains instead of clinical isolates [51,52,53]. The innovative aspect of our study is the use of clinical isolates. In our research, the best antibacterial activity was demonstrated by the cinnamon EO, which effectively inhibited all clinical isolates of *P. aeruginosa*, and among them, almost half reacted at a very low MIC (0.0125% *v*/*v*).

Previous research described some antimicrobial mechanisms of action, which were attributed to the main chemical constituents of each EO. For example, in the case of cinnamon, studies show trans-cinnamaldehyde to be the main constituent [54]. Many articles describe the antimicrobial activity of trans-cinnamaldehyde [55,56], but the mechanism of action of trans-cinnamaldehyde is not completely elucidated [57]. Some describe mechanisms of action, such as the inhibition of ergosterol synthesis in fungi or the capacity to inhibit *Escherichia coli* adhesion to the human epithelial cells [58,59]. In our case, besides trans-cinnamaldehyde, which was also the main aromatic compound, a significant amount of coumarin was also present. Plants with large amounts of coumarin are of interest to the scientific community due to their biological benefits. Besides the antibacterial effects of coumarin, other studies prove the anti-tumoral, anti-coagulant, or even anti-inflammatory activity with real potential benefits [60,61]. Our results fill this gap with proof related to the inhibition of RNA synthesis and the deregulation of gene expression in bacterial cells after only a short exposure to very small concentrations of cinnamon EO.

Linalool, another compound with antimicrobial activity found in basil (47.66% in our results), has shown positive results on species like *Pseudomonas*, *Klebsiella,* or *Listeria* in some studies. Our study has found only 5 out of 72 clinical isolates of *P. aeruginosa* to be inhibited by basil EO at a low MIC [62,63,64,65].

Oregano is described in many studies as having a very efficient antimicrobial activity due to the presence of carvacrol [66,67], which damages bacterial cells using permeabilization and depolarization of the cytoplasmic membrane. Interestingly, in our study, oregano EO did not show inhibitory effects on *P. aeruginosa* clinical isolates. [68,69,70]. The combination formulations of EOs (OnGuard© and DDR Prime©) showed no antimicrobial effects on our clinical *P. aeruginosa* isolates, as opposed to results of other studies. For example, OnGuard© was shown to present an inhibitory effect on *P. aeruginosa*, but the experiment was performed on a standard strain [71]. These discrepancies further support the importance of testing the bioactive effects of EOs on clinical isolates rather than on standard strains (which, on the other hand, are valuable for method reproducibility reasons).

In our opinion, it is possible to assume that the antibacterial activity of the EO is due to the combination of the compounds, not necessarily only due to the main compound. The inhibition of efflux pumps might have a crucial role in the management of the treatment. The activity of EOs with potential inhibitory activity over the efflux pumps, concomitant with modern antibiotics administration, might influence the activity of the treatment with different results. For example, *Croton zehntneri*, as an inhibitor of the *NorA* efflux pump, increased the norfloxacin antimicrobial activity on *Staphylococcus aureus* by almost 40% [72,73]; or *Thymus maroccanus* and *Thanasis broussonetia* EOs inhibited efflux pumps in Gram-negative bacteria, which lead to increased susceptibility to chloramphenicol [74].

The inhibition of *mex*E and *mex*Y gene expression by using *Satureja khuzistanica* EO was presented in recent research, affecting the MexEF-oprM and mexXY-oprM efflux pumps activity [75]. These results are comparable with those from our study, where some *mex* genes were under-expressed in the presence of cinnamon EO. Our study also showed over-expression of other *mex* efflux pump genes after exposure of *P. aeruginosa* strains to cinnamon EO, and this can be due to the bacterial effort to eliminate the harmful agents out of the bacterial cell [76]. Nevertheless, bacteria were not able to efficiently pump out the bioactive compounds, as our results demonstrated a significant decrease in RNA quantity, overall inhibition of gene expression after three hours of contact with cinnamon EO, and bacterial death after 16–18 h of contact with cinnamon EO, as proved by MIC and MBC.

## 4. Materials and Methods

### 4.1. Bacterial Strains

During the routine laboratory testing, all *P. aeruginosa* strains identified from the patients admitted to the Mures County Clinical Hospital Romania (MCCH) between 2020–2022 were isolated and conserved by freezing at −80 °C in Tryptic Soy Broth (TSB) with 10% glycerin, for further use. All the bacterial isolates were identified using classical methods (culture characters, positive oxidase test, growth on cetrimide) and tested for antibiotic resistance using Kirby–Bauer disk diffusion and confirmed using MIC testing on Vitek 2 System (Biomerieux, France). Antibiogram results were interpreted following the EUCAST 2022 standard. The study was approved by the Ethical Board of MCCH (no. 15190/19.10.2020).

The inclusion criteria for the working group: strains that showed to be MDR, XDR, or PDR—these correspond to resistance in at least one antibiotic from three or more antibiotic classes, susceptibility to only one or two antibiotic classes, respectively resistance to all antibiotic classes [77]. All bacteria also had to present resistance to carbapenems. Following the inclusion criteria, a number of 72 strains were used for further testing.

### 4.2. Essential Oils (EOs)

A total of 16 EOs were selected to be tested for their antibacterial effects on *P. aeruginosa*. Fourteen were pure EOs, and 2 were combined formulations (mixed EOs of different plant families, officially manufactured by the producer) (Table 3). All the EOs were ordered directly from the manufacturer that guarantees the best quality pure products. Each EO was accompanied by a quality certificate showing the HPLC analysis performed by the producer, which could be freely downloaded from www.sourcetoyou.com, based on each EO lot number present on the essential oil bottles. The chemical analysis was not performed for the combined formulations of EOs.

### 4.3. Antimicrobial Activity of EOs

The antimicrobial activity of EOs was assessed in two steps. First, a disk diffusion method was used as a screening test in order to establish which of the 16 selected EOs presented potential antimicrobial activity on the clinical isolates of *P. aeruginosa*. MIC testing was performed for the EO that proved antibacterial activity following the disk diffusion screening. The methods were adapted according to EUCAST guidelines regarding the antibiotic susceptibility testing documents [78,79].

#### 4.3.1. Disk Diffusion Method

Mueller Hinton Agar plates (Oxoid, Holdings Ltd., Altrincham, United Kingdom) were inoculated with 0.5 McFarland suspensions of each *P. aeruginosa* isolate. After drying, 3 blank paper disks (Oxoid, Holdings, Altrincham, United Kingdom) of 6 mm each were placed on the surface of one agar plate and loaded with 10 μL of each EO. Disks of Meropenem 10 μg (Oxoid, Holdings, Altrincham, United Kingdom) served as the control for each isolate. A blank disk (negative control) was also placed on the plate, proving its sterility and absence of any inhibitory side effects. The plates were incubated at 35 °C for 18 h, and the inhibition zone diameters were measured. Isolates that showed diameters ranging between 6–10 mm were considered resistant to the EO activity and were not tested further. For all the strains that presented diameters of more than 10 mm, a MIC test was performed for the corresponding EO. Meropenem diameters <24 mm confirmed the resistance/low susceptibility of the tested strains to carbapenems. A control strain of *P. aeruginosa* ATCC 27853 and Meropenem 10 μg were used to validate the disk diffusion methodology.

#### 4.3.2. Minimum Inhibitory Concentration (MIC)

The MIC was assessed using microdilution method in sterile 96-well plates. For this, each EO had to be sequentially diluted in water to obtain decreasing concentrations. As the EO cannot be directly mixed in water, it was solubilized as follows: 100 μL of EO was mixed with 1 μL of Tween20 and 100 μL of sterile distilled water, heated in a thermomixer at 50 °C for 10 min, and then sonicated for 10 min at 45 Hz, in a sonicator water bath at 25 °C (Elmasonic S30, Elma Schmidbauer GmbH, Singen, Germany).

In the meantime, 100 μL of sterile distilled water was distributed in the 96-well plates in columns 2–12 using a multichannel micropipette. The prepared EO (200 μL) was distributed in the first column of each 96-well plate. From the first column, 100 μL of EO solution was sequentially transferred to the next columns, obtaining serial dilutions. From the last column, the excess of 100 μL was discarded.

A 0.5 McFarland (2 × 10^8^ colony forming units/mL) bacterial inoculum was prepared in sterile saline from fresh cultures of *P. aeruginosa* on CLED (Cystine Lactose Electrolyte Deficient) agar (Oxoid, Holdings ltd, Ireland). Of this, 10 μL were mixed with 9990 μL of Mueller Hinton Broth (MHB) 2X (Oxoid, Holdings ltd, Ireland), and 100 μL of this mixture was transferred over the EO dilutions. Thus, the final concentrations of EO in the plate columns are 25%, 12.5%, 6.25%, 3.13%, 1.56%, 0.78%, 0.39%, 0.20%, 0.10%, 0.05%, 0.025%, and 0.0125% *v*/*v*. A positive control consisting of serially diluted Tween20 in water and bacterial inoculum in MHB 2x, without added EO, was prepared for each tested strain. A negative control (water, MHB 2x, and 1% Tween20) was also used for each plate to prove sterility.

The plates were incubated at 35 °C in the normal atmosphere for 16–18 h. The MIC was considered in the last well of each row where no visible bacterial growth was observed and was interpreted as *v*/*v* percentage of stock solution.

#### 4.3.3. Minimum Bactericidal Concentration (MBC)

From the last 3 wells of each row of the MIC plate that showed no bacterial growth, 3 μL were inoculated on Sheep Blood Agar, labeled in a checkerboard pattern, being able to identify the correspondence of each well. The plates were incubated at 35 °C in the normal atmosphere for 18 h. The MBC was considered the position with no bacterial growth and interpreted as *v/v* percentage of stock solution.

### 4.4. Gene Expression of the Efflux Pumps

The gene expression of efflux pumps was determined for 15 randomly selected *P. aeruginosa* isolates, with and without exposure to cinnamon EO (which proved to have the best antibacterial effect following the MIC and MBC tests).

#### 4.4.1. Bacterial RNA Extraction

From fresh cultures of each *P. aeruginosa* prepared on Sheep Blood Agar, one colony was inoculated in 4 mL TSB (Oxoid Ltd., Thermo Fisher, Heysham, UK) in sterile 2 mL microcentrifuge tubes and incubated 18 h at 35 °C.

The gene expression of efflux pumps was assessed in the presence of MIC concentrations of *Cinnamomum zeylanicum* EO. For this, 1 mL TSB containing MIC concentration of EO (%*v*/*v*) was incubated for 3 h at 35 °C to allow the EO to exert the effects. After the incubation time, the tubes were centrifuged at 12,000× *g* for 2 min, the supernatant was carefully discarded, and the deposit was washed with sterile saline to remove the traces of the EO. The bacterial RNA was extracted with Quick RNA Midiprep Kit (Zymo Research, Irvine, CA, USA) following the manufacturer protocols, obtaining 35 μL of RNA.

For each *P. aeruginosa* isolate, the RNA was also extracted in the absence of EO (gene expression control).

The quantity of RNA obtained was evaluated using spectrophotometry (BioPhotometer D30, Eppendorf AG, Hamburg, Germany).

#### 4.4.2. DNase Treatment

Prior to the reverse transcription, in order to digest the eventual traces of DNA, 15 μL of the extracted RNA were treated with one unit of RNase-free DNase I enzyme (Thermo Scientific, Vilnius, Lithuania) according to manufacturer protocols.

#### 4.4.3. Reverse Transcription

For all extracts, the RNA concentration was adjusted with DNase-free water, so an amount of 300 ng RNA was to be included in the reverse transcription. The reaction was performed using GoScript Reverse Transcription Kit (Promega, Madison, WI, USA). Each reaction was performed in a master mix containing 4 μL of random primers, 2 μL of an enzyme, and 4 μL of DNase-free water. The RNA obtained for the DNase treatment was added to the master mix solution obtaining a final reaction volume of 25 μL.

The recommended protocol used for the cDNA synthesis was: one cycle at 25 °C for 5 min, 42 °C for 60 min, followed by 70 °C for 15 min.

#### 4.4.4. RT-PCR (Real-Time Polymerase Chain Reaction)

RT-PCR was performed in order to evaluate the gene expression of carbapenemase efflux pumps in the presence and absence of the EO. The five primer pairs, specific for *mex*A, *mex*B, *mex*C, *mex*E, and *mex*X efflux pumps (Table 4), were selected from existing literature and modified where needed to obtain the same melting temperature [80]. Housekeeping gene specific primers for *P. aeruginosa* (*rpo*D) were also selected in order to evaluate the ΔΔCT value (Table 4) [80,81].

The PCR was performed using GoTaq^®^ qPCR Master Mix, respecting the manufacturer protocol (10 μL qPCR Master Mix 2X, 1 μL of 10 μM forward primer, 1 μL of 10 μM reverse primer, 0.2 μL CXR (Carboxy-X-Rhodamine) as passive reference, 6.8 μL water, 1 μL cDNA). The final volume of the mix was 20 μL, with primer concentration adjusted at 0.5 μM. The qPCR protocol consisted of one cycle of initial denaturation at 95 °C for 2 min, followed by 40 cycles of 2-step amplification (95 °C for 2 min and annealing/extension at 60 °C for 1 min), followed by a melting curve.

## 5. Conclusions

There is sustainable evidence for the beneficial effects of EOs, especially considering the outstanding inhibition of bacterial growth using *Cinnamomum zeylanicum* EO. Definitely, each bacterial strain is unique, as our results show that *P. aeruginosa* clinical isolates, part of the same species, with similar resistance phenotype, tested in the same conditions, present different individual responses to the same natural antibacterial agent. The great potential of cinnamon EO to be used as an adjuvant to modern antibiotic treatments is supported by its bactericidal effects and its ability to interfere with bacterial gene expression/RNA synthesis and modulate the efflux pump activity. It is important to continuously develop and discover new alternative ways for limiting bacterial infections in the era of multidrug antibiotic resistance.

## Figures and Tables

**Figure 1 antibiotics-12-00163-f001:**
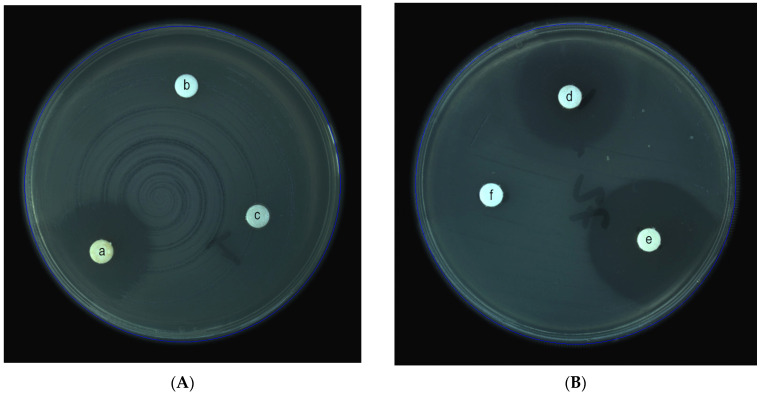
(**A**,**B**) Representative images for Kirby–Bauer disk diffusion screening results, showing the zones of inhibition: (a,e) *Cinnamomum zeylanicum*; (b,c,f) *Origanum vulgare*, *Melaleuca alternifolia*, respectively *Curcuma longa*; (d) *Thymus vulgaris*.

**Figure 2 antibiotics-12-00163-f002:**
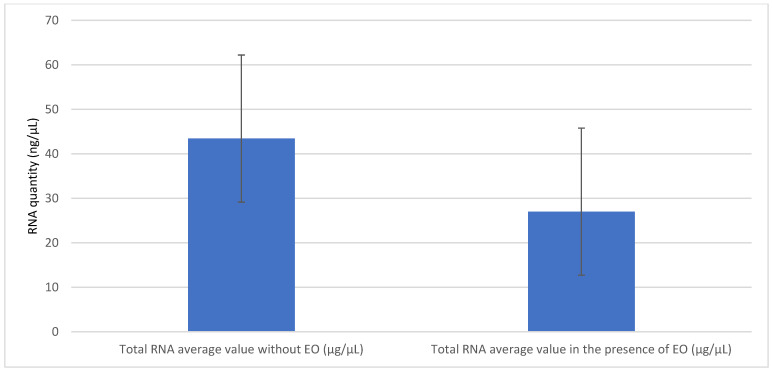
The comparison of average values of total RNA after extraction, with or without exposure to the cinnamon EO.

**Figure 3 antibiotics-12-00163-f003:**
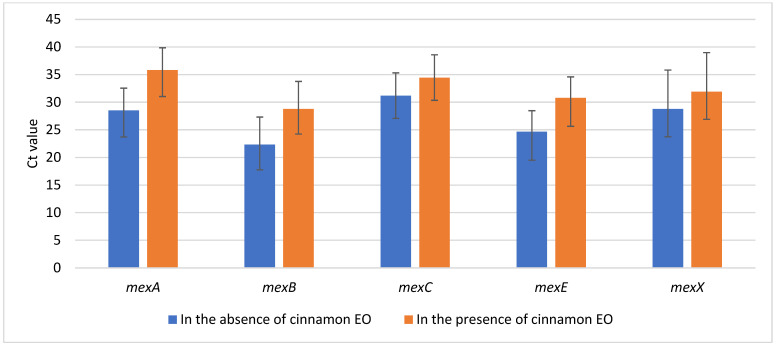
The comparison of average values of Ct in the presence and absence of cinnamon EO. The error bars represent the variability of Ct values among the 15 tested isolates using real-time RT-PCR.

**Table 1 antibiotics-12-00163-t001:** The main chemical compounds described in the tested EOs.

Essential Oils	Chemical Composition	%
Cinnamon	trans-cinnamaldehyde	55.14%
	trans cinnamyl acetate	11.97%
	β-phellandrene	5.19%
Basil	linalool	47.66%
	1,8-cineole	10.2%
	trans-α-bergamotene	5.82%
Clove	eugenol	80.43%
	eugenyl acetate	12.56%
	β-caryophyllene	5.16%
Hawaiian Sandal	cis-α-santalol	43.21%
	cis-β-santalol	18.14%
	cis-lanceol	8.24%
Lavender	linalool	34.24%
	linalyl acetate	30.49%
	lavandulyl acetate	4.89%
Lemon eucalyptus	Citronellal	72.96%
	Citronellol	10.09%
	neo-Isopulegol	4.66%
Marjoram	terpinen-4-ol	24.86%
	γ-terpinene	13.92%
	trans-sabiene-hydrate	12.73%
Tea tree	terpinen-4-ol	37.65%
	γ-terpinene	19.64%
	α-terpinene	10.44%
Oregano	carvacrol	65.19%
	para-cymene	8.65%
	thymol	8.47%
Patchouli	patchouli alcohol	38.04%
	α-bulnesene	16.28%
	α-guaiene	11.71%
Peppermint	menthol	35.45%
	menthone	25.63%
	menthyl acetate	6.74%
Rosemary	1,8-cineole	45.27%
	α-pinene	12.69%
	camphor	10.83%
Turmeric	ar-turmerone	38.25%
	α-turmerone	12.32%
	α-curcumene	5.1%
Thyme	thymol	33.03%
	para-Cymene	24.92%
	γ-terpinene	14.09%

**Table 2 antibiotics-12-00163-t002:** Summary of MIC testing on clinical isolates of *P. aeruginosa*.

EO	MIC (% *v*/*v*)												
	Total Samples (% of Total No. of Isolates)	25	12.5	6.25	3.13	1.56	0.78	0.39	0.2	0.1	0.05	0.025	0.0125
Cinnamon	72 (100%)	-	-	-	-	-	-	-	-	-	30.55% *n* = 22	33.33% *n* = 24	47.22% *n* = 34
Thyme	27 (37.5%)	88.88% *n* = 24	3.7% *n* = 1	-	-	3.70% *n* = 1	-	3.70% *n* = 1	-	-	-	-	-
Turmeric	1 (1.38%)	-	100% *n* = 1	-	-	-	-	-	-	-	-	-	-
Peppermint	4 (5.55%)	75% *n* = 3	-	-	-	-	-	-	25% *n* = 1	-	-	-	-
Basil	6 (8.33%)	16.66% *n* = 1	-	-	-	-	-	33.33% *n* = 2	50% *n* = 3	-	-	-	-
Clove	6 (12.5%)	100% *n* = 6	-	-	-	-	-	-	-	-	-	-	-
Lavender	9 (12.5%)	55.55% *n* = 5	-	11.11% *n* = 1	-	11.11% *n* = 1	-	11.11% *n* = 1	-	-	11.11% *n* = 1	-	-

**Table 3 antibiotics-12-00163-t003:** List of EO tested for anti-*Pseudomonas* activity.

	Plant Family	EO Plant Species (Lot Number)
**Pure EO** (***n* = 14**)	*Lamiaceae*	*Rosmarinus officinalis* (Rosemary) 2019711Y)
		*Origanum majorana* (Marjoram) (2017512Y)
		*Thymus vulgaris* (Thyme) (190219Y)
		*Ocimum basilicum* (Basil) (201414Y)
		*Pogostemon cablin* (Patchouli) (202842Y)
		*Origanum vulgare* (Oregano) (192497Y)
		*Lavandula angustifolia* (Lavender) (212161Y)
		*Mentha piperita* (Peppermint) (211651Y)
	*Myrtaceae*	*Melaleuca alternifolia* (Tea tree) (2025211Y)
		*Eucalyptus citriodora* (Lemon eucalyptus) (213505Y)
		*Eugenia caryophyllata* (Clove) (201748Y)
	*Lauraceae*	*Cinnamomum zeylanicum* (Cinnamon) (211124Y)
	*Santalaceae*	*Santalum paniculatum* (Hawaiian sandal) (202469Y)
	*Zingiberaceae*	*Curcuma longa* (Turmeric) (2034311Y)
**Combined EO formulations** (***n* = 2**)	Combination A (OnGuard©)	*Capparis mitchellii* (Wild orange) *Eugenia caryophyllata* (Clove) *Cinnamomum zeylanicum* (Cinnamon) *Eucalyptus citriodora* (Lemon eucalyptus) *Rosmarinus officinalis* (Rosemary)
	Combination B (DDR Prime©)	*Myrtus communis* (Myrtle) *Capparis mitchellii* (Wild orange) *Litsea cubeba* (Mountain pepper) *Thymus vulgaris* (Thyme) *Eugenia caryophyllata* (Clove) *Melaleuca quinquenervia* (Niaouli) *Philadelphus coronarius* (Mock-orange)

**Table 4 antibiotics-12-00163-t004:** The presentation of the efflux pump primers.

Efflux Pump Gene	Primer Sequence (5′ > 3′)	Amplicon Length bp (Base Pair)
*mex*A-Fw	ACCTACGAGGCCGACTACCAGA	252 bp
*mex*A-Rw	GTTGGTCACCAGGGCGCCTTC	
*mex*B-Fw	GTGTTCGGCTCGCAGTACTCGA	244 bp
*mex*B-Rw	AACCGTCGGGATTGACCTTGAGC	
*mex*C-Fw	ACGTCGGCGAACTGCAACG	374 bp
*mex*C-Rw	AGCCAGCAGGACTTCGATACCG	
*mex*E-Fw	TCATCCCACTTCTCCTGGCGC	151 bp
*mex*E-Rw	CGTCCCACTCGTTCAGCGG	
*mex*X-Fw	CCAGCAGGAATAGGGCGACCA	82 bp
*mex*X-Rw	AATCGAGGGACACCCATGCACATC	
*rpo*D-Fw	GCGGATGATGTCTTCCACCTGTTCC	132 bp
*rpo*D-Rw	GCGCAACAGCAATCTCGTCTGAAAGA	

## Data Availability

Data sharing not applicable.

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
