# Peer review of "Antibacterial Effect of 16 Essential Oils and Modulation of mex Efflux Pumps Gene Expression on Multidrug-Resistant Pseudomonas aeruginosa Clinical Isolates: Is Cinnamon a Good Fighter?"

_antibiotics, 2023, doi:10.3390/antibiotics12010163_

Round 1
Reviewer 1 Report
The manuscript Antibacterial effect of 16 essential oils and modulation of mex 2 efflux pumps gene expression on multidrug-resistant Pseudomonas aeruginosa clinical isolates: is cinnamon a good fighter? Has been reviewed. Some errors and mentioned below. However, a thorough revision is needed for the manuscript. The manuscript is not up to the standard and needs considerable revision before publication.
1. Line no. 17 – remove subheading -Material and methods
2. Line no. 22 – remove subheading -Results:
3. Line No. 102-103- The chemical composition of the 14 essential oils, as detected by HPLC, are presented in Table 1. ??? were tested through the Kirby-Bauer disk diffusion method- rewrite the sentence.
4. The essential oils are usually diluted in antimicrobial studies. Here the authors used directly in the disc. Please discuss why it is directly used. Without an emulsifier or diluting whether the essential oil will diffuse through the medium?
5. HPLC analysis data are missing even in the supplementary file. Further please explain the identification procedure of the chemical components.
6. Line 131- ‘in vivo’ should be in Italics
7. Line 139- Also, no antimicrobial inhibitory effects were found in the case of mixed EO formulations A and B. Methodology for this result is not clear. Did you perform a checkerboard assay?
8. All abbreviations need a full form when used for the first time in the manuscript.
9. The introduction needs to add the significance of efflux pump genes (especially mex 2 efflux pumps gene).
10. Conclusion doesn’t contain any information about the efflux pump genes or the outcomes study related to gene expression.
11. Grammatical and typographical errors should be corrected.
Author Response
Thank you for the valuable comments which made the manuscript significantly improved! We hope that following our point-by-point reply, the submission can be accepted for publication.
- Line no. 17 – remove subheading -Material and methods
Response: Done
- Line no. 22 – remove subheading -Results:
Response: Done
- Line No. 102-103- The chemical composition of the 14 essential oils, as detected by HPLC, are presented in Table 1. ??? were tested through the Kirby-Bauer disk diffusion method- rewrite the sentence.
Response: Thank you, it was a part of a phrase that remained by mistake in this line.
- The essential oils are usually diluted in antimicrobial studies. Here the authors used directly in the disc. Please discuss why it is directly used. Without an emulsifier or diluting whether the essential oil will diffuse through the medium?
Response: For screening purposes only, we consider that the potential antimicrobial effects can be assessed by the disk-diffusion assay, using the essential oils in their crude state, without any solubilizing agents, according to the many already published studies
In vitro Susceptibility Testing of Essential Oils against Gram-positive and Gram-negative Clinical Isolates, including Carbapenem-resistant Enterobacteriaceae (CRE) | Open Forum Infectious Diseases | Oxford Academic
Antibacterial activity and interactions of plant essential oil combinations against Gram-positive and Gram-negative bacteria - ScienceDirect
Moreover, the fact that cinnamon exerted an antimicrobial effect on all strains, proves that the methodology is reliable.
- HPLC analysis data are missing even in the supplementary file. Further please explain the identification procedure of the chemical components.
Response: The HPLC was performed by the producer, as part of their quality control of the commercialized EO, and it was stated in the methods section. We have also added a supplemental file with the HPLC analysis, and we modified the results to be more transparent: “The primary chemical components of the 14 EO are presented in Table 1. The primary chemical components of the 14 EO are presented in Table 1, according to the HPLC (High Performance Liquid Chromatography) analysis that was performed by the producer during the quality control protocol. ”
- Line 131- ‘in vivo’ should be in Italics
Response: Done
- Line 139- Also, no antimicrobial inhibitory effects were found in the case of mixed EO formulations A and B. Methodology for this result is not clear. Did you perform a checkerboard assay?
Response: No, it is not the case. The mixed EO formulations are official products from the same producer, namely OnGuard, respectively DDR Prime. In chapter 4.2 we mentioned that “2 were combined formulations (mixed EO of different plant families, officially manufactured by the producer)”.
- All abbreviations need a full form when used for the first time in the manuscript.
Response: Thank you for letting us know, we modified as needed
- The introduction needs to add the significance of efflux pump genes (especially mex 2 efflux pumps gene).
Response: Thank you for the suggestion, we added more information about the role of mex pumps.
- Conclusion doesn’t contain any information about the efflux pump genes or the outcomes study related to gene expression.
Response: The last paragraph mentions the inhibitory effect of cinnamon EO on efflux pumps: „There is a huge potential for cinnamon EO, but not limited to, due to its bactericidal effects, inhibitory effect on efflux pumps, or as a general gene expression/RNA inhibitor, to be used as an adjuvant to the modern antibiotic treatments”. We would not like to repeat the results in the conclusion section. If the reviewer strongly suggests, we agree to add more information in conclusions on these findings.
- Grammatical and typographical errors should be corrected.
Response: Thank you, the manuscript was carefully revised and many errors were corrected.
Author Response
Thank you for the valuable comments which made the manuscript significantly improved! We hope that following our point-by-point reply, the submission can be accepted for publication.
Abstract:
- According to the standard format of the journal there should be no use of sub headings such as; Background, methods, results and conclusion.
Response: Done
- Write the full name without abbreviations when it is used for the first time in the text e.g. XDR and PDR in line #29.
Response: Done
- There is no clear hypothesis provided in the abstract portion. The abstract portion is mainly provided with study lay out, a brief methodology but no information of control.
Response: Thank you, we mentioned the controls also in the abstract
- There are previous report available on the cinnamon as antimicrobial agent against Pseudomonas aeruginosa such as:
- https://www.ncbi.nlm.nih.gov/pmc/articles/PMC6474160/
- https://link.springer.com/article/10.1186/s12906-016-1134-9
- https://www.sciencedirect.com/science/article/abs/pii/S0926669015305598
Then what is the novelty of this research work. Explain the novelty statement.
Response: Thank you, we agree that many publications are available, regarding the antibacterial effects of EO against bacteria. Nevertheless, our study was performed on MDR/PDR/XDR clinical isolates. Other studies used carbapenem-resistant isolates only, or ATCC strains. Moreover, our study assessed (beside the antibacterial effects), the interference of EO with the RNA messaging system of these isolates.
Introduction:
- Italicize the biological name of species in line # 35 according to binomial nomenclature.
Response: Done
- Write the full name without abbreviations when it is used for the first time in the text CDC in line # 49.
Response: Done
- According to standard format of the journal there must be use of comma between the multiple citations which must be ensured instead of hyphen in line # 57, 67, 83 and 85.
Response: The references were formatted with Zotero bibliography manager software, based on the template provided by MDPI. Also, the MDPI guide mdpi_references_guide_v5.pdf (mdpi-res.com) suggests using hyphens for multiple citations.
- This section is provided with old literature therefore, needs up to date references.
Response: Thank you for the suggestion, in some situations, the older references are defining some of the terms, described for the first time after 2010. We have indeed updated some of the references as suggested, so it results that 60% of the introduction references are no older than 4 years.
- The title of the article reflects essential oil and bacterial species Pseudomonas aeruginosa and the abstract reflects Kirby-Bauer disk diffusion method but this section lack related information to them.
Response: Thank you, we have updated the introduction with details on the diffusion methods used in the detection of carbapenem resistance.
Results:
- Write the full name without abbreviations when it is used for the first time in the text e.g. XDR in line # 98.
Response: Done.
- The paragraph sentence from (From the 160 strains of… (SD=8.73 mm) is not clear and do not give any sense e.g. what the author means “stoked from previous research” if this is link to any previous findings or any other previous work done why not provided with proper reference???
Response: Thank you, the isolates were stocked for research purposes in our lab. We changed the phrase as follows: “From the 160 strains of P. aeruginosa stoked during 2020-2022 in our laboratory, 72 corresponded to the inclusion criteria of highly increased resistance”.
- The presentation of data in table 1 is highly unprofessional from the sentence “chemical composition of the 14 essential oils, as detected by HPLC” in line number 102. In general it revealed that the composition of oils was performed by the author through HPLC. This is against the sentence mentioned in methodology (section 4.2) in line # 324-327. If all the information is available at the given link then why need to provide table 1.
Response: Thank you for noticing this. Indeed, the HPLC was performed by the producer, as part of their quality control of the commercialized EO. In completion to the methods section which presents this, we have also modified the results, as suggested: “The primary chemical components of the 14 EO are presented in Table 1, according to the HPLC (High Performance Liquid Chromatography) analysis that was performed by the producer during the quality control protocol. ”
- There is no consistency in the wording of sentence in line # 109-111. It is better to write (for thyme, 8.33% (n=6) …. and for peppermint to turmeric 1.38% (n=1).
Response: Thank you, indeed it was unclear. We modified as follows: “For the rest of EO, various results were obtained: P. aeruginosa presented susceptibility to thyme (37.5% of all isolates; n=27), clove (8.33%; n=6), lavender (12.5%; n=9), basil (8.33%; n=6), peppermint (5.55%; n=4) and turmeric (1.38%; n=1). All the other EO failed to prove any antibacterial effect on all P. aeruginosa isolates, showing no inhibition zone following the disk-diffusion method.”
- The author has given information related to “meropenem” but this section has no information about the negative control used in the experiment.
Response: Thank you, indeed we omitted to add information about the control
- Italicize the biological name of species in line # 143 and 145 according to binomial nomenclature.
Response: Thank you, we modified all the names with biological ones, for better uniformity.
- There is no continuity among the wordings of result section specially section of gene expression. I therefore suggest to the author to make sections and sub sections of the results according to the materials and methods section.
Response: Thank you, we have modified as suggested.
- The quality of figure 2 and 3 is extremely poor.
Response: Thank you, the figures were recreated in a lossless format.
- The last three values of MIC for Cinnamon in table 2 are ambiguous. As going down the concentration activity must be decreases as mentioned it is concentration dependant. Then why at low dose of concentration the activity is higher than the activity at high dose of concentration???
Response: the strains used in this study are clinical multiresistant strains, isolated from different patients. Thus, we have to expect that their behavior to the EO activity can be also different. Some presented MIC at 0.05%, other at 0.025% and the majority at 0.01%. Anyway, all strains were inhibited at concentrations of <= 0.05% of cinnamon EO.
Discussion
- According to standard format of the journal citation must be provided in number and not by name. This must be ensured in line # 184, 197, 274 and 292.
Response: thank you, we have modified according to the requirements.
- According to standard format of the journal there must be use of comma between the multiple citations which must be ensured instead of hyphen in line # 210, 244, 267 and 271.
Response: The references were formatted with Zotero bibliography manager software, based on the template provided by MDPI. Also, the MDPI guide mdpi_references_guide_v5.pdf (mdpi-res.com) suggests to use hyphen for multiple citations.
- Merge the small paragraphs from line # 201 to 211 and line # 279 to 285
Response: Thank you, done.
- This section is provided with a pattern exactly like a review article as most of the discussion based on previous findings, it lacks a concise discussion and needs to be rewrite according to the standard format of the journal.
Response: Thank you, indeed the details made the section hard to be lectured. We rewrote most of the discussions in a more condensed form, to reflect our findings.
Material and methods
- Providing information merely confirmation of HPLC results with provided link is not sufficient. Proper information such as retention time and chromatogram and other HPLC conditions must be provided, where the EO lot numbers are??? If all the information is available at the given link then why need to provide table 3.
Response: Thank you, the lot numbers were added in Table 3. Due to copyright issues, we cannot republish the HPLC documents, but these can be accessed freely as stated in materials and methods.
- This section is provided with reproducible methodology however lack any reference of previous protocol in most parameters.
Response: Thank you, references regarding the methodology were added: 50, 77-81.
- I repeat here as stated in the result portion; this portion reflects positive control but lack information to negative control.
Response: Thank you, we added the information
- This section is provided with different sections such as 4.1, 4.2, 4.3 and 4.4 but to attract the audience section 4.3 and 4.4 needs sub sections.
Response: Thank you, we have numbered the subsections as suggested.
Conclusion:
Conclusion section is unnecessary lengthy. The author should focus on the results/findings and future recommendations. Therefore this section needs to be rephrasing.
Thank you, the conclusion was rephrased in a more concise way, as suggested.
References:
This section provided according to the standard format of the journal and is okay. Reference # 54 is missing the publication year.
Response: Thank you, we have rechecked all the references to be according to the standards.
Reviewer 3 Report
The study be Răzvan Lucian Coșeriu et al have investigated the antibacterial activity of 16 common essential oils (EO) on multidrug-resistant (MDR) Pseudomonas aeruginosa. Following that , the effects on mex efflux pumps gene expression have investigated . Based on the data obtained there is no doubt on the results of this study . The manuscript contains a piece of work that is sufficiently and interesting. All aspects of the manuscript are complete. However, there is some minor issued need to address before consider this manuscript for publication. There is inconsistency or not clear information on the number of EO that have been tested . In the abstract author have mentioned that 16 common essential oils have tested . Then , in the results section only 14 have mentioned and as in material and ,method section. Please check it.
Line 143 and 145 : please write bacteria name in italic
Line 150: Figure 2 . is not clear . please use high quality figure
Line 256: Please add full stop after the sentence “beside the effects desc”
Author Response
Thank you for the valuable comments which made the manuscript significantly improved! We hope that following our point-by-point reply, the submission can be accepted for publication.
There is inconsistency or not clear information on the number of EO that have been tested . In the abstract author have mentioned that 16 common essential oils have tested . Then , in the results section only 14 have mentioned and as in material and ,method section. Please check it.
Response: We have tested 14 pure EO, plus 2 combined formulations (OnGuard and DDR Prime). Table 3 was reformatted to better present all the EO.
Line 143 and 145 : please write bacteria name in italic
Response: Thank you, we have modified.
Line 150: Figure 2 . is not clear . please use high quality figure
Response: Thank you, the figures were recreated in a lossless format.
Line 256: Please add full stop after the sentence “beside the effects desc”
Response: Thank you for noticing, we repaired the phrase.
Round 2
Reviewer 1 Report
The authors clarified almost all the comments raised by the reviewer. The manuscript has been improved and can be accepted for publication.
